# How Does Industrial Upgrading Affect Carbon Productivity in China's Service Industry?

**Shimei Weng and Jianbao Chen ***

College of Mathematics and Statistics, Fujian Normal University, Fuzhou 350117, China;
wengshimei1995@126.com
* Correspondence: jbjy2006@126.com

**Abstract:** Promoting carbon productivity is an effective way to reduce carbon emissions. The existing literature focuses mainly on the carbon productivity of heavily polluted sectors, such as heavy industry, the manufacturing industry, and the construction industry. With the deepening of China's economic transformation and industrial upgrading, the service industry plays an increasingly important role in the national economy, and the ratio and amount of carbon emissions in the service industry show an upward trend. In order to effectively achieve the goal of energy conservation and emission reduction, it is necessary to study how industrial upgrading affects the carbon productivity in the service industry. This study uses a spatial autoregressive panel model to investigate the carbon productivity in China's service industry. The empirical results are summarized as follows: (1) the carbon productivity of China's service industry is on the rise, and there exist regional heterogeneity and spatial dependence; (2) industrial upgrading has a significant positive effect on the carbon productivity in China's service industry; (3) the positive effect of industrial upgrading in the eastern (northern) region is higher than that in middle and western (southern) regions in the service industry; and (4) environmental regulation and economic development have positive moderating effects in the process of industrial upgrading. Accordingly, some targeted policy suggestions are put forward.

**Keywords:** industrial upgrading; carbon productivity; spatial autoregressive panel model; regional heterogeneity; moderating effect

## 1. Introduction

Tackling global climate change has become a world consensus. In 2015, the Paris Agreement set a long-term target increase in global average temperature of within 2 °C relative to the pre-industrial levels. However, the global average temperature increase reached 1.2 °C in 2020 [1]. This means that all countries should make great efforts to reduce carbon dioxide ($CO_2$) emissions and achieve the goal of carbon peak and carbon neutrality as soon as possible. As the second largest economy, China is the world's highest $CO_2$ emitter [2]. In 2021, China's primary energy consumption and $CO_2$ emissions accounted for 26.49% and 31.06% of the global share, respectively [3]. Therefore, China is facing more severe pressure on global climate governance and greenhouse gas emission reduction.

The service sector has long been considered a "clean" industry, with low energy consumption and emissions; thus, China's emission reduction policies have been focused mainly on traditional energy-intensive industries [4–6]. However, with the rapid development of the service economy, the service industry has become the main force driving in China's economic growth and new energy consumption. According to China's Industrial Classification of National Economy (GB/T 4754-2017), the service industry includes producer services, such as transportation and information transmission industry, as well as life service industries, such as accommodation, retail, and education industry. The role of the service industry in China's economy has become increasingly prominent, and its industrial added value has grown rapidly from 2000 to 2020. As shown in Figure 1, the output of

the service industry increased from CNY 3989.91 billion in 2000 to CNY 24,173.33 billion in 2020 (constant prices in 2000), with an average annual growth rate of 9.43%. During this period, the energy consumption of the service industry rose from 375.11 million ton coal equivalent (Mtce) in 2000 to 1471.05 Mtce in 2020; $CO_2$ emissions increased from 529.54 million tons (Mt) to 1303.34 Mt [7]. In terms of the world economy as a whole, the final energy consumption of the industrial sector was 120.27 million trillion joules (TJ), of which China accounted for 37.35%; the residential sector's consumption was 88.52 million TJ, 17.28% of which was accounted for by China; and the commercial and public service sector's consumption was 31.75 million TJ, with China accounting for 11.53% (2020 data) [8]. Figure 1 shows that China's industrial sector accounts for a relatively high proportion of the world's energy consumption, while its world share of energy consumption in the service sector has increased significantly. In fact, high energy consumption brings high carbon emissions. Among many sub-industries, the direct carbon emission level of productive service sectors, such as transportation, storage, and electronic information is very high, which is closely related to their high energy consumption. Conversely, the lifestyle service sectors, such as hotels and catering, medical care, and social security, absorb most of the emissions through an industrial chain transfer, making a significant contribution to indirect carbon emissions [9]. These facts mean that the process of the development of the service economy is not "green" and "pollution-free" [10], and pollution emissions caused by the development of the service industry cannot be ignored. Simultaneously, with the rising cost of carbon emission reduction, the reduction potential of the industrial sector will gradually decrease in the future [11–13]. As a result, the service industry provides an alternative direction for further promoting energy conservation. Hence, improving the carbon productivity is of great significance to fully tap the emission reduction potential of the service industry and promote low-carbon development.

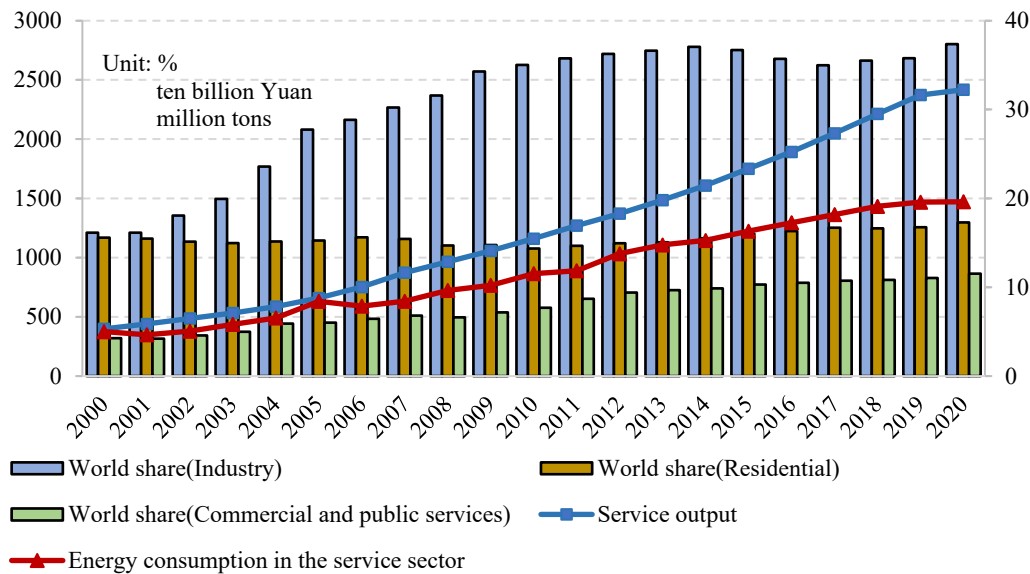

**Figure 1.** Development status of China's service industry.

Identifying key drivers of the carbon productivity in the service sector is essential for effective environmental protection policies. Existing research shows that industrial upgrading is a critical factor affecting carbon emissions and an effective way to transform the mode of economic development [14]. Industrial upgrading can change the existing energy consumption pattern, allocate and utilize resources reasonably, and effectively reduce carbon emissions. Studying the impact of industrial upgrading on carbon productivity is important for fully exploiting China's carbon emission reduction potential.

Therefore, by using China's interprovincial panel data from 2005 to 2019, this study examines the impact of industrial upgrading on the service industry's carbon productivity

and the moderating role of environmental regulation and economic development on the above relationship. Studies show that industrial upgrading has a positive impact on carbon productivity, and this impact can occur through three main mechanisms: adjusting the proportion among industries, promoting technological innovation and energy-saving technologies, and optimizing the industrial structure for green development. In addition, it can be positively adjusted through environmental regulation and economic development, and this impact shows obvious heterogeneity across geographical locations.

This study makes three contributions. First, it expands the research perspective. Most prior studies focused on carbon productivity in traditional industries. However, recent research suggests that carbon emissions in the service sector cannot be ignored [15]. Therefore, this study extends the research of regional carbon productivity from primary and secondary industries to the service industry and reveals its spatial–temporal characteristics and influence mechanisms. This contributes to achieving emission reduction goals in the service industry. Second, we take full account of the inter-regional and inter-temporal interdependence of the observations and use spatial econometrics to analysis the impact of industrial upgrading. This dramatically reduces the error caused by the traditional econometric method. Third, it enriches the boundary mechanism and regional heterogeneity of the industrial upgrading affecting carbon production in the service sector. Considering the synergistic effect of the environment and the economy in industrial development, we discuss the regulatory role of environmental regulation and economic development. This will provide a more effective theoretical support for the implementation of the policies.

## 2. Literature Review

Carbon productivity, proposed by Kaya and Yokobori (1997) [16], is a valid indicator to evaluate the efficiency of a green economy [17]. It combines carbon emission reduction with stable economic growth and fully reflects the win–win requirements of growth and the environment [18].

The existing literature conducts systematic research on carbon productivity and other related indicators from three main aspects.

(1) Regional differences and industry heterogeneity. Liu et al. (2019) [19] studied regional differences in carbon emissions. Their study found a declining trend in carbon emissions across China's regions, but higher emissions in the western area. Moreover, due to the unique attributes of the industry, there are differences in the energy demand in different sectors, as well as considerable differences in carbon productivity and emissions. Xu et al. (2019) [20] discovered that the industrial sector has the highest proportion of carbon emissions in all industries. Schäfer and Yeh (2020) [21] pointed out that the transportation industry plays a critical role in carbon reduction. Therefore, scholars have extended the research perspective on carbon emissions and carbon productivity to industries such as industry, transportation, power, agriculture, and tourism. For example, the industrial carbon productivity in Chinese provinces has been assessed by Long et al. (2016) [22]. In their recent papers, Wu et al. (2018) [23] and Xie et al. (2019) [24] explored the decoupling of carbon emissions with the GDP in the Chinese power sector and construction industry, respectively.

(2) Influencing factors and driving forces. In the existing literature, the technological progress [25–28], economic development level [29], industrial structure [30–32], environmental regulation [33], urbanization level [34], and energy structure [35,36] are considered to be important factors influencing carbon productivity. Many scholars used exponential decomposition, scenario simulation, and traditional econometric methods to examine the factors affecting the emissions, intensity, efficiency, and productivity. Wang et al. (2020) [37] studied the drivers of $CO_2$ emissions in the country with the second largest carbon emitter (the United States) based on the LMDI decomposition model. They found that the scale effect was an important influencing factor, while the technology effect was the key driver. Chen et al. (2022) [38] applied the spatio-temporal LMDI method and found that the GDP per capita and the indus-

trial structure are the most crucial factors on urban emissions. Taking the EU power industry as an example, Karmellos et al. (2021) [39] applied a decomposition model to explore the seven drivers of $CO_2$ discharge. Based on the scenario modeling approach, Ang and Goh (2019) [40] found that differences in carbon intensity originate mainly from the electricity sector, and the proportion of coal has the greatest impact. Zhao et al. (2022) [41] found that increased energy tax regulation and sci-tech investment can effectively reduce carbon intensity. Song and Han (2022) [42] pointed out that market-based environmental regulation measures could help improve carbon productivity, while control-based regulation policies may have the opposite effect. However, in the study of Hu and Wang (2020) [43], appropriate regulatory policies had obvious effects on carbon productivity. In addition, Wang et al. (2018) [44] regarded economic growth as a crucial factor based on a time series model, and Li and Wang (2019) [45] explored the improvement plan of carbon productivity from the perspective of a socio-economic development strategy.

(3) The relationship between industrial upgrading and carbon productivity-related indicators. With the acceleration of industrial transformation, increasingly more literature has confirmed the crucial role of the industrial structure in China's mitigation. However, the issue of whether industrial structure optimization can efficiently reduce $CO_2$ has not been unified. For example, Ahmed and Zeshan (2014) [46] found that industrial restructuring can effectively reduce energy consumption. The analysis of Yang et al. (2022) [47] showed that upgrading an industry's carbon performance can be enhanced by enhancing its efficiencies. However, Gu et al. (2016) [48] believed that the industrial structure upgrade contributed little to carbon mitigation.

In sum, in the context of global climate change, many scholars have launched intense discussions on the relevant content of the carbon field. However, the existing research still has shortcomings. (1) Existing studies on carbon productivity focus mainly on the primary and secondary sectors, and they rarely involve the service sector. However, there is no denying that the service industry's carbon emissions affect the environment. In particular, with the increasing share of the service sector and the diminishing marginal contribution from the traditional sectors to carbon reduction, the service sector is more likely to become the main force in the future carbon reduction process. Therefore, studying carbon productivity in the service sector has important practical significance. (2) Most current studies do not consider the spatial properties of carbon productivity. They generally assume that the variables themselves are independent of each other in the traditional econometric methods, but this assumption is difficult to attain in reality. Wang et al. (2021) [49] pointed out that inter-regional carbon emissions are spatially correlated, and carbon emissions among adjacent provinces may affect each other. This conclusion also applies to energy consumption [50] and carbon productivity [51]. Accordingly, we speculate that the service sector's carbon productivity in neighboring regions may also have a spatial correlation. In this scenario, traditional econometric models cannot take into account the spatial correlation among observations, which may induce bias in the estimation results [52]. (3) Currently, industrial upgrading is not well studied in the existing literature, and few studies examine how it affects carbon productivity in the service sector. Furthermore, given the regional differences in economic foundations, the impact of industrial upgrading may not be fully convergent across regions, and less attention is paid to this issue in the current research.

## 3. Theoretical Hypotheses

### 3.1. The Impact of Industrial Upgrading

At present, with the increasingly prominent status of the service industry, it has become particularly imperative to promote its low-carbon economic growth. An industrial structure adjustment is the fundamental way to reduce carbon emissions, which also plays a unique role in the service industry's carbon productivity. Industrial upgrading can promote carbon productivity in the service industry in three ways: First, industrial upgrading can affect carbon productivity in services by adjusting the proportion among the industries.

Compared with the traditional industries, the tertiary industry has the advantages of lower energy dependence, higher economic efficiency, and less environmental pollution [53]. Therefore, industrial upgrading and transformation leads to a gradual increase in the service sector's proportion, and the corresponding shares of the other two industries gradually decline. Meanwhile, the service industry continues to absorb the labor and capital factors that initially belonged to the traditional industries [54], which eventually increases the economic scale and carbon productivity. Secondly, industrial upgrading plays its role by enhancing the allocation of resource elements among various industries and within the service industry. Industrial adjustment can optimize the allocation of multiple production factors among industries [55], optimize the full utilization of the resources in various departments within the services, improve energy efficiency [56], effectively restrain the expansion of carbon emissions and accelerate the growth of the service industry output [46]. Therefore, the carbon productivity in services increases accordingly. Finally, industrial upgrading acts on carbon productivity by driving the progress of low-carbon technologies. Industrial upgrading inevitably leads to the elimination of some low-end industries, while technology-intensive industries with good development prospects and green and low-carbon characteristics develop rapidly [57]. In addition, it contributes to developing low-carbon technologies, enabling the service sector to improve its carbon productivity. Accordingly, we put forward the first hypothesis.

**H1.** *Industrial upgrading positively affects the carbon productivity in the service sector.*

### 3.2. Heterogeneity of Industrial Upgrading Affects Carbon Productivity in the Service Industry

Owing to differing economic fundamentals and geographical characteristics, there are regional differences in the industrial level and carbon productivity in the service sector [19]. The specific effect and direction on the service sector's carbon productivity in other regions also may not be identical. Generally speaking, areas with relatively weak economic foundations and lower industrial levels may lag behind the process of improving the service economic scale. Furthermore, due to the limited technical level, it is difficult for all sectors in the region to rejuvenate. For regions with a solid economic foundation and high industrial level, the emission-reduction technologies of various industrial sectors have become mature [58], and the accelerated pace of industrial upgrading effectively improves carbon productivity. In addition, different energy demands affected by geographical location and climate characteristics may also induce regional heterogeneity of the industrial upgrading effect. Accordingly, we formulate the second hypothesis.

**H2.** *There is regional heterogeneity in the impact of industrial upgrading on carbon productivity in the service sector.*

### 3.3. The Moderating Effect

As a comprehensive social policy, environmental regulation is an effective means for local governments to regulate enterprise activities for the governance of regional environments [59]. China's environmental regulation policy has gone through the evolution process of exploration stage, buffer stage, and optimization and innovation stage, and the policy tools have been continually enriched [60]. At the current stage, an environmental policy pattern with the coexistence of a market-oriented mechanism and command-control mechanism has been formed [61]. Hence, appropriate environmental regulation policies are highly likely to influence the industrial structure and carbon productivity of the service industry [62]. Simultaneously, under different policy intensities, the role of industrial upgrading on productivity of the service industry may not be entirely consistent. The continual increase in regulation policies will be helpful for the formation of access standards for enterprises in various industries. Under strong policy constraints, industrial upgrading has accelerated the elimination or transformation of some high-energy-consuming and low-efficiency enterprises [63]. The service industry enterprises with comparative advantages have developed more rapidly, and the scale of benefits in the service industry

has been previously expanded. In addition, it can stimulate technological innovation in the service industry enterprises [64]. Through the "innovation compensation effect", a strong impetus will be injected into industrial upgrading and, thus, promote the enterprises' mitigation ability.

All along, economic development and industrial upgrading are mutually reinforcing. Therefore, it is reasonable to infer that economic development will adjust the impact of industrial upgrading. It is largely due to the optimized allocation of resources in China's growth, which has strengthened the advantages of industrial upgrading. The rapid economic growth makes the allocation of internal resource elements in the service industry more reasonable, and the increasingly optimized economic development level and the industrial structure will be beneficial to the carbon productivity in the service industry. In addition, economic development can bring more preferential policies and subsidies to the modern service industry. Good national social policies will further promote the agglomeration of industry and human capital and directly drive the low-carbon technological progress in services. This also provides an impetus for the low-carbon upgrading, thereby enhancing the positive effect of upgrading. Therefore, we put forward the third and fourth assumptions as follows:

**H3.** *Environmental regulation plays a positive moderating role between industrial upgrading and service carbon productivity.*

**H4.** *Economic development plays a positive moderating role between industrial upgrading and service carbon productivity.*

The above analysis shows that industrial upgrading contributes to carbon productivity in services, and environmental regulation and economic development can regulate the influence of industrial upgrading. Figure 2 illustrates the mechanism.

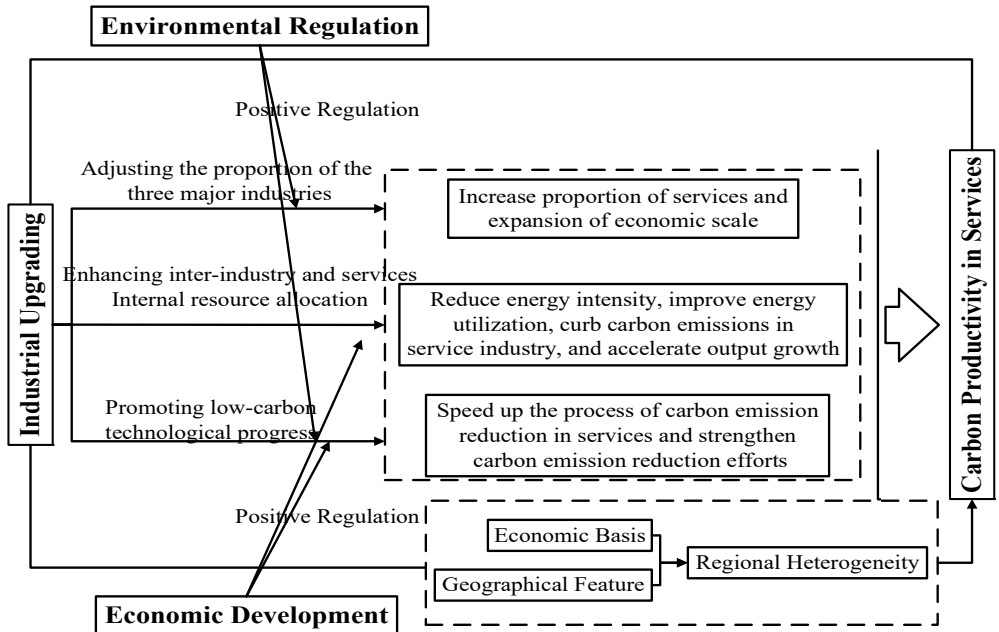

**Figure 2.** The mechanism of industrial upgrading, environmental regulation, and economic development on carbon productivity in services.

## 4. Methodology

### 4.1. Model Setting

Due to the close relationship among the social economies in China's different regions, the service sector's carbon productivity shows a specific spatial dependence, that is, regions in adjacent locations in a certain spatial dimension have similar values. Considering that

traditional panel models ignore the existence of spatial effects and the estimation results may have an extensive bias or even obtain the exact opposite conclusion, while spatial panel models can avoid this problem to a large extent, we first test the spatial autocorrelation. The specific mathematical expressions are as follows:

$$I = \frac{\sum\limits_{i=1}^{N}\sum\limits_{j=1}^{N} w_{ij}(CPS_i - \overline{CPS})(CPS_j - \overline{CPS})}{s^2 \sum\limits_{i=1}^{N}\sum\limits_{j=1}^{N} w_{ij}} \tag{1}$$

$$C = \frac{(N-1)\sum\limits_{i=1}^{N}\sum\limits_{j=1}^{N} w_{ij}(CPS_i - \overline{CPS})^2}{2(\sum\limits_{i=1}^{N}\sum\limits_{j=1}^{N} w_{ij})[\sum\limits_{i=1}^{N} (CPS_i - \overline{CPS})^2]} \tag{2}$$

where $CPS_i$ denotes the service sector's carbon productivity in the $i$—th region, and $\overline{CPS}$ and $s^2$ are the mean and standard deviation of carbon productivity, respectively. $w_{ij}$ is the $ij$—th element in the spatial weight matrix. In addition, $I \in [-1, 1]$, $C \in [0, 2]$, $0 < I < 1$, and $0 < C < 1$ all indicate positive spatial autocorrelation for carbon productivity in services, and vice versa for negative spatial autocorrelation.

This study uses mainly the spatial lag model (SAR) and the spatial error model (SEM) to analyze the results. Regarding the specific selection of these two models, we conduct LM and robust LM tests on the residual items estimated by OLS [65] and screen the two models according to the test results. The basic formulas are shown below.

SAR Model:

$$\begin{aligned} \ln CPS_{it} &= \alpha_0 + \rho W \ln CPS_{it} + \alpha_1 \ln IS_{it} + \alpha_2 \ln ER_{it} + \alpha_3 \ln TI_{it} + \alpha_4 \ln URB_{it} \\ &+ \alpha_5 \ln ED_{it} + \alpha_6 \ln EL_{it} + \alpha_7 \ln FD_{it} + u_i + \varepsilon_{it} \end{aligned} \tag{3}$$

SEM Model:

$$\begin{aligned} \ln CPS_{it} &= \beta_0 + \beta_1 \ln IS_{it} + \beta_2 \ln ER_{it} + \beta_3 \ln TI_{it} + \beta_4 \ln URB_{it} + \beta_5 \ln ED_{it} \\ &+ \beta_6 \ln EL_{it} + \beta_7 \ln FD_{it} + u_i + \varepsilon_{it} \\ \varepsilon_{it} &= \lambda W \varepsilon_{it} + \mu_{it} \end{aligned} \tag{4}$$

where subscript $i$ is the $i$—th region and $t$ is the $t$—th time period. $\alpha_0$ and $\beta_0$ are intercept terms. Both $\alpha_1 \sim \alpha_7$ and $\beta_1 \sim \beta_7$ represent the regression coefficients of the independent variables. $\rho$ and $\lambda$ represent the spatial lag and the spatial error coefficient, respectively. $W$ is the spatial weight matrix. $u_i$ is the $i$—th individual fixed effect. $\varepsilon_{it}$ and $\mu_{it}$ represent the error terms.

*4.2. Variable Selection*

4.2.1. Carbon Productivity in Services

Carbon productivity in the service industry (CPS) is the explained variable, and it refers to the ratio of the added value in the service industry to its $CO_2$ emissions. The specific calculation formula is

$$CPS_{it} = \frac{Y_{it}}{C_{it}} \tag{5}$$

where $CPS_{it}$, $Y_{it}$, and $C_{it}$ represent carbon productivity level, the added value, and the $CO_2$ emissions, respectively, in services in the $t$ year of the region $i$. The value-added data of the service industry are deflated to the value-added data of the base period of 2005 according to the "third industry value-added index". The calculation of $CO_2$ emissions in services is collated mainly using provincial data from the China Carbon Accounting Database (CEADs) for 2005–2019 [66,67].

### 4.2.2. Industrial Upgrading

Industrial upgrading (*IS*) is the core explanatory variable. Among all production activities, the secondary industry's development is critical to economic growth, but it also leads to large consumption of energy. In contrast, the $CO_2$ emissions of the tertiary industry are relatively low. Generally speaking, with industrial upgrading, the carbon productivity in the service industry has also been effectively improved. This study refers to Wu and Liu (2021)'s [68] calculation for industrial upgrading:

$$\theta_q = \arccos\left(\frac{X_0 X_q}{\|X_0\| \cdot \|X_q\|}\right), \ q = 1, 2, 3 \tag{6}$$

$$IS = \sum_{k=1}^{3} \left(\sum_{q=1}^{k} \theta_q\right) \tag{7}$$

where $\theta_q$ is the vector angle, $X_0$ is a spatial vector composed of the ratio of the three industries, $X_q$ is the unit vector, and *IS* is the industrial upgrading calculated by the vector angle $\theta_q$.

### 4.2.3. Other Variables

Environmental regulation (*ER*) and economic development (*ED*) are moderator variables. *ER* is measured by the proportion of total investment in environmental pollution control. It can force enterprises to bring about the application of cleaner technology, which will help alleviate environmental problems. *ED* is expressed by the actual per capita GDP, and the data are reduced using 2005 as the base period.

Considering that carbon productivity in the service industry may also be influenced by other economic and social variables, we introduced the following control variables.

Technological progress (*TI*). Technological progress is conducive to a low-carbon economy, thereby improving the service sector's carbon productivity. We select the percentage of R&D expenditures in GDP to represent it.

Urbanization (*URB*). Urbanization (measured by the ratio of urban population) has two effects on carbon emissions in services. On the one hand, the acceleration of urbanization has attracted more talent to the cities and supported the development of the service industry. On the other hand, a large number of population surges and spatial expansion may also increase the level of energy consumption. All these may affect carbon emissions in the service industry.

Education level (*EL*). Education level has a continuous increasing effect. Improving the education level will drive the comprehensive improvement of labor quality, thus positively affecting carbon productivity in the service sector. *EL* is measured by the percentage of the employed population relative to the education level (high school and above).

Financial development (*FD*). Financial development also has two effects: one is to curb energy consumption in the service industry, and the other is to promote foreign direct investment in the service sector, resulting in a "pollution haven" effect. Therefore, we select the proportion of total deposits and loans of financial institutions at the end of the year to measure the level of financial development.

In addition, in the following robustness test, this paper also presents workforce size (*LB*) and energy consumption (*ES*) for further control, which are represented by the number of employees in the service industry and the total energy consumption of the service industry, respectively.

### 4.2.4. Data Description

We select 30 provincial panel datasets in China from 2005 to 2019 (excluding Tibet, Taiwan, Hong Kong, and Macau because of missing data) for analysis. These data are obtained from the "China Statistical Yearbook", "EPS Global Statistical Database", "China Energy Statistical Yearbook", "China Tertiary Industry Statistical Yearbook", and the annual

statistical yearbook of various provinces. All data containing prices are deflated to constant prices using a base period of 2005, and most of the variables are taken in natural logarithmic form to eliminate the heteroscedasticity of the data (Table 1).

**Table 1.** Descriptive statistics of variables.

| Variable | Meaning | Mean | Standard Deviation | Min | Max |
| --- | --- | --- | --- | --- | --- |
| ln *CPS* | Carbon productivity in services | 5.1954 | 0.5101 | 3.9052 | 6.4725 |
| ln *IS* | Industrial upgrading | 1.8848 | 0.0470 | 1.7773 | 2.0349 |
| ln *ER* | Environmental regulation | 0.1839 | 0.4809 | −1.2069 | 2.2212 |
| ln *TI* | Technological progress | −1.9080 | 0.9818 | −3.6348 | 1.1163 |
| ln *URB* | Urbanization | 3.9670 | 0.2521 | 3.2910 | 4.4954 |
| ln *ED* | Economic development | 10.3090 | 0.6718 | 8.5275 | 12.0163 |
| ln *EL* | Education level | 3.3348 | 0.4351 | 2.1412 | 4.3883 |
| ln *FD* | Financial development | 1.0087 | 0.3277 | 0.2532 | 2.0957 |
| ln *LB* | Workforce size | 6.5476 | 0.8259 | 4.4751 | 8.2025 |
| *ES* | Energy consumption | 0.1622 | 0.1027 | 0.0157 | 0.5702 |

## 5. Results

### 5.1. China's Service Sector Carbon Productivity and Its Spatial Distribution Characteristics

Figure 3 displays the spatial distribution of provincial carbon productivity levels in 2005 and 2019. First, China's overall service industry carbon productivity showed an upward trend, rising from 15,753 CNY/ton in 2005 to 29,249 CNY/ton in 2019, and the overall carbon productivity in services was effectively improved. Compared with 2005, carbon productivity increased significantly. Especially in the eastern areas, carbon productivity exceeded 45,000 CNY/ton by 2019, and the carbon productivity in Zhejiang reached 64,700 CNY/ton. However, carbon productivity in the midwest areas increased slowly. By 2019, Qinghai and Guizhou were both still in the low-value range, and the carbon productivity was less than 15,000 CNY/ton. Overall, the regional distribution of the service industry's carbon productivity did not change significantly from 2005 to 2019, and the carbon productivity showed a stepped distribution from the eastern to the western regions. In addition, Figure 3 also illustrates the similarity of the carbon productivity among adjacent provinces. Therefore, we assumed that the service industry's carbon productivity had a spatial autocorrelation, which needed further testing.

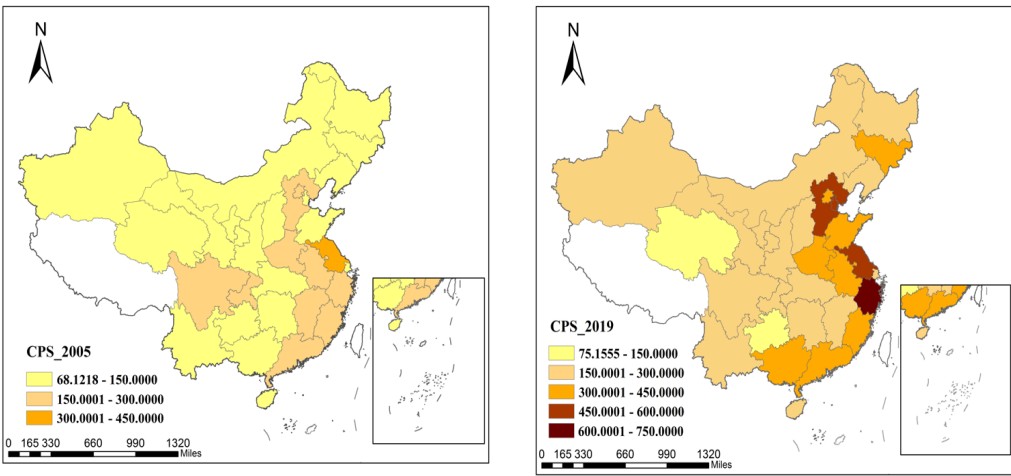

**Figure 3.** Distribution of service sector's carbon productivity.

### 5.2. Spatial Autocorrelation Test

The carbon productivity in the service industry passed the spatial autocorrelation test, in which Moran's I was significantly higher than the expected value of 0 (Table 2). At the

same time, Geary's C was considerably lower than the expected value of 1. The service sector's carbon productivity had a high degree of spatial dependence. In addition, the temporal trends of Moran's I and Geary's C also showed an inverse movement, which was in line with reality. Moran's I generally indicated an increasing trend, and the space centralization effect of the service sector's carbon productivity in different regions was gradually increasing.

**Table 2.** Spatial autocorrelation test results.

| Year | Moran's I | Z-Value | *p*-Value | Geary's C | Z-Value | *p*-Value |
|------|-----------|---------|-----------|-----------|---------|-----------|
| 2005 | 0.069 *** | 3.154 | 0.001 | 0.921 ** | −1.743 | 0.041 |
| 2006 | 0.065 *** | 3.073 | 0.001 | 0.924 * | −1.601 | 0.055 |
| 2007 | 0.064 *** | 3.039 | 0.001 | 0.926 * | −1.495 | 0.067 |
| 2008 | 0.057 *** | 2.809 | 0.002 | 0.936 * | −1.354 | 0.088 |
| 2009 | 0.060 *** | 2.920 | 0.002 | 0.937 * | −1.313 | 0.095 |
| 2010 | 0.059 *** | 2.885 | 0.002 | 0.935 * | −1.358 | 0.087 |
| 2011 | 0.063 *** | 2.993 | 0.001 | 0.932 * | −1.412 | 0.079 |
| 2012 | 0.057 *** | 2.831 | 0.002 | 0.933 * | −1.390 | 0.082 |
| 2013 | 0.088 *** | 3.734 | 0.000 | 0.898 ** | −2.269 | 0.012 |
| 2014 | 0.100 *** | 4.090 | 0.000 | 0.890 *** | −2.468 | 0.007 |
| 2015 | 0.105 *** | 4.242 | 0.000 | 0.883 *** | −2.635 | 0.004 |
| 2016 | 0.103 *** | 4.199 | 0.000 | 0.885 *** | −2.517 | 0.006 |
| 2017 | 0.108 *** | 4.349 | 0.000 | 0.886 *** | −2.525 | 0.006 |
| 2018 | 0.114 *** | 4.512 | 0.000 | 0.889 *** | −2.509 | 0.006 |
| 2019 | 0.108 *** | 4.346 | 0.000 | 0.890 *** | −2.412 | 0.008 |

Note: ***, **, and * denote variables significant at 1%, 5%, and 10% significance levels, respectively; the tables below are the same.

### 5.3. Basic Regression Results

Before carrying out the primary empirical research, we conducted a panel unit root test on the eight variables ln *CPS*, ln *IS*, ln *ER*, ln *TI*, ln *URB*, ln *ED*, ln *EL*, and ln *FD* involved in the text. The results indicated that all variable series were stationary. In order to determine the specific fitting model, our study considered two cases (with and without control variables). According to Table 3 and the discriminant criteria of Anselin et al. (2004) [65], we believe that the selection of the SAR model was appropriate. In addition, we fitted the SAR and SEM models for the two scenarios, respectively. The model fit results (Table 3) show that in both cases, the $R^2$ and LogL of the SAR model were better overall than those of the SEM model. Meanwhile, the Hausman test lent support to the fixed effects. Therefore, the following was fit to the SAR model under the fixed effects to study the related issues. Furthermore, we also verified the consistency of the main results using stepwise regression. It is worth mentioning that the results also showed that the SAR model should be used in various situations where the control variables are gradually introduced.

According to Table 3, the overall spatial panel model fit the sample data better ($R^2$ and LogL are relatively high). The spatial coefficients of all models were significantly positive, indicating that the service industry's carbon productivity levels in various regions of China were spatially dependent. That is, provinces with high (low) carbon productivity in services can be affected by those in the surrounding areas. To specifically describe the influence of each explanatory variable on service sector's carbon productivity, this study analyzed mainly the SAR model results after introducing all the control variables (Table 3).

The impact coefficient of *IS* on *CPS* in services was significantly positive. This result was supported by the SAR model with no control variables and the gradual introduction of the control variables, which indicated that industrial upgrading was propitious in improving the service industry's carbon productivity; thus, H1 is established. Since different industries have great differences in $CO_2$ emissions and social benefits, the service industry requires less energy and consumes less energy, resulting in huge economic benefits and relatively high carbon productivity. With the advanced development of the industry, production factors, such as labor capital, will gradually be transferred to the service sector, and

the economic scale and benefits of the service industry will increase accordingly. Industrial upgrading can increase service output while reducing $CO_2$ emissions by enhancing the efficiency of the resource allocation among industries and within the service industry and improving utilization efficiency. Furthermore, the upgrading and adjustment of industrial designs can also promote the progress of technology-intensive industries, thereby improving the green technology levels in services and reducing carbon emissions.

**Table 3.** Spatial panel model regression results.

| Variable | No Control Variable | | Step-by-Step Introduction of Control Variables | | | | | Introduce All Control Variables | |
|---|---|---|---|---|---|---|---|---|---|
| | **SAR** | **SEM** | **SAR (1)** | **SAR (2)** | **SAR (3)** | **SAR (4)** | **SAR (5)** | **SAR** | **SEM** |
| ln *IS* | 2.1660 *** (3.99) | 1.0505 (1.28) | 2.1595 *** (4.07) | 1.8180 *** (3.35) | 1.9702 *** (3.05) | 1.2881 ** (1.96) | 1.2925 ** (1.97) | 1.3223 * (1.93) | 0.9812 (1.30) |
| ln *ER* | | | −0.0940 *** (−4.24) | −0.0902 *** (−4.12) | −0.0866 *** (−3.70) | −0.0821 *** (−3.60) | −0.0833 *** (−3.65) | −0.0831 *** (−3.64) | −0.0800 *** (−3.37) |
| ln *TI* | | | | 0.1244 *** (3.55) | 0.1230 *** (3.50) | 0.0819 ** (2.35) | 0.0815 ** (2.34) | 0.0833 ** (2.25) | 0.0808 ** (2.15) |
| ln *URB* | | | | | −0.0510 (−0.43) | −0.7267 *** (−4.42) | −0.6516 *** (−3.63) | −0.6480 *** (−3.58) | −0.6513 *** (−3.26) |
| ln *ED* | | | | | | 0.3038 *** (5.72) | 0.3273 *** (5.67) | 0.3273 *** (5.67) | 0.5498 *** (7.84) |
| ln *EL* | | | | | | | −0.0694 (−1.04) | −0.0687 (−1.02) | −0.0957 (−1.21) |
| ln *FD* | | | | | | | | −0.0132 (−0.15) | 0.0358 (0.37) |
| $\rho/\lambda$ | 0.6688 *** (11.03) | 0.8145 *** (21.63) | 0.6771 *** (11.41) | 0.6587 *** (10.82) | 0.6680 *** (10.49) | 0.5428 *** (7.03) | 0.5423 *** (7.03) | 0.5446 *** (6.94) | 0.5895 *** (7.45) |
| $R^2$ | 0.5551 | 0.4714 | 0.5451 | 0.5558 | 0.5509 | 0.5920 | 0.5897 | 0.5895 | 0.5695 |
| LogL | 177.8320 | 169.1829 | 186.6461 | 192.8881 | 192.9803 | 209.2364 | 209.7720 | 209.7832 | 207.3736 |
| LM-Error | 9.979 [0.002] | | 10.753 [0.001] | 10.547 [0.001] | 7.292 [0.007] | 5.251 [0.022] | 4.789 [0.029] | 2.500 [0.114] | |
| Robust LM-Error | 4.543 [0.033] | | 4.944 [0.026] | 4.742 [0.029] | 2.484 [0.115] | 1.645 [0.200] | 1.458 [0.227] | 0.304 [0.581] | |
| LM-Lag | 77.662 [0.000] | | 86.287 [0.000] | 87.775 [0.000] | 98.117 [0.000] | 96.237 [0.000] | 94.022 [0.000] | 87.923 [0.000] | |
| Robust LM-Lag | 72.226 [0.000] | | 80.477 [0.000] | 81.97 [0.000] | 93.308 [0.000] | 92.632 [0.000] | 90.692 [0.000] | 85.728 [0.000] | |

Note: ***, **, and * denote variables significant at 1%, 5%, and 10% significance levels, respectively; Values in parentheses are Z-values, and those in square brackets are *p*-values.

Environmental regulation negatively inhibited *CPS* in the overall fierce competition in the domestic service industry. Internal enterprises are subject to strict environmental constraints in the process of seeking their own development, which makes enterprise operation more difficult. Some enterprises may take negative measures to reduce or even stop production, and the overall economic benefits of the service industry will be reduced. In addition, service-oriented enterprises invest a large amount in sewage treatment fees and environmental protection fees in the production process, which undoubtedly increase the production burden and crowd into other profitable investments. This also leads to a decline in capital returns, which is harmful to *CPS*, and the "following cost effect" was obvious.

In the model, technological innovation can promote *CPS*. First of all, technological innovation helps to speed up the internal production process of service firms, improve the labor production efficiency and economic development efficiency, and subsequently increase the service sector's output. In addition, technological innovation can improve green energy-saving technologies, which will reduce the emissions in services.

The coefficient of urbanization affecting carbon productivity in services was −0.648. For every 1% increase in urbanization, the carbon productivity decreased by 0.648%. With the advancement of urbanization, services such as transportation directly or indirectly increased energy consumption and hindered the improvement of carbon productivity.

Economic development had an obvious pulling influence on *CPS*. This is mainly because regions with better economic development tend to have greater financial advantages, and the regional service industry will pay more attention to applying energy-saving technology in the process of development. As a result, less carbon dioxide was emitted

while creating huge economic benefits, so the regional carbon productivity in services was relatively high. In addition, education level and financial development did not affect the carbon productivity of the regional service industry.

*5.4. Robustness Test*

5.4.1. Deleting the Last Year Sample

To examine the research conclusions' robustness, we deleted the sample data of the last year (2019) and re-fitted the SAR model to estimate the deleted sample data (column (1) of Table 4). The empirical results, such as industrial upgrading, still positively affected the carbon productivity in services, and only the significance of some variables changed, but it did not affect the overall research interpretation. Accordingly, the above research results have a certain robustness.

**Table 4.** Robustness test results.

| Variable | Remove Last Year Sample | Substitute Core Explanatory Variables | Add Control Variables | Replace Spatial Weight Matrix |
|---|---|---|---|---|
| ln *IS* | 1.6147 ** | 1.2353 ** | 1.0292 * | 2.2594 *** |
| | (2.07) | (2.19) | (1.93) | (3.27) |
| ln *ER* | −0.0911 *** | −0.0822 *** | −0.0738 *** | −0.0917 *** |
| | (−3.69) | (−3.60) | (−4.17) | (−3.88) |
| ln *TI* | 0.0720 * | 0.0838 ** | 0.0572 ** | 0.0843 ** |
| | (1.88) | (2.27) | (1.97) | (2.20) |
| ln *URB* | −0.6437 *** | −0.6321 *** | −0.7662 *** | −0.5903 *** |
| | (−3.40) | (−3.52) | (−5.45) | (−3.13) |
| ln *ED* | 0.3525 *** | 0.3291 *** | 0.5129 *** | 0.4078 *** |
| | (5.92) | (5.71) | (10.48) | (7.01) |
| ln *EL* | −0.0647 | −0.0671 | 0.064 | −0.0777 |
| | (−0.97) | (−1.00) | (1.22) | (−1.12) |
| ln *FD* | −0.117 | −0.0315 | −0.0494 | 0.0295 |
| | (−1.13) | (−0.35) | (−0.72) | (0.32) |
| ln *LB* | | | 0.1068 * | |
| | | | (1.92) | |
| *ES* | | | −3.0671 *** | |
| | | | (−17.13) | |
| $\rho$ | 0.4887 *** | 0.5257 *** | 0.6666 *** | 0.1917 *** |
| | (5.47) | (6.42) | (10.95) | (3.51) |
| $R^2$ | 0.5528 | 0.5959 | 0.7380 | 0.5854 |
| LogL | 208.1892 | 210.3180 | 321.9140 | 197.2386 |

Note: ***, **, and * denote variables significant at 1%, 5%, and 10% significance levels, respectively; Values in parentheses are Z-values.

5.4.2. Replacing the Core Explanatory Variable

The core explanatory variable, industrial upgrading, has many alternative indicators, and they generally highlight the important position in the service industry in the three major industries. Therefore, considering that the research results may be contingent and one-sided, the following indicators were constructed to re-measure industrial upgrading by referring to the practice of Xu and Jiang (2015) [69] and Zheng et al. (2021) [70]. The specific construction method is as follows:

$$IS' = 1 \times q_1 + 2 \times q_2 + 3 \times q_3 \tag{8}$$

where $q_i$ $(i = 1, 2, 3)$ represents the share of the $i$—th industry in the GDP.

After constructing new industrial upgrading variables, the original core explanatory variables were replaced, and the SAR model under fixed effects was fitted again for estimation (Table 4). It showed that both $R^2$ and LogL of the model increased, indicating that the model fit was better than before. At the same time, industrial upgrading, technological innovation, and economic development continued to contribute significantly to improv-

ing the carbon productivity in services, while environmental regulation and urbanization significantly negatively affected the carbon productivity in services, and human capital and financial development had no evident impact. These are in full agreement with the previous empirical results, which once again proved their reliability.

### 5.4.3. Adding Control Variables

To fully ensure the stability of the impact of industrial upgrading and to explore the effects of other factors as much as possible, this study further controlled two variables that were closely related to the service sector on the basis of Model (3), including energy consumption and the size of the labor force in the service sector. Table 4 shows that the results of industrial upgrading were consistent with the above, which proved the stability of the conclusions. In addition, energy consumption and workforce size produced effective negative and positive effects, respectively. This is because the increase in the size of the labor force in the service industry rapidly expanded the economic scale, while the increase in energy consumption led to the increase in carbon emissions, which, in turn, led to the realistic impact on *CPS* in the service industry.

### 5.4.4. Replacing the Spatial Weight Matrix

Our research further confirmed the robustness by replacing the inverse distance geospatial weight matrix $W_g$ with the adjacency spatial weight matrix $W_a$ and reusing the SAR model for regression. The significance and direction of all variables were basically the same as the estimated results of the SAR model fitted after introducing all control variables in Table 3; only the coefficients and significance levels were different.

### 5.5. Endogenous Processing

Due to the possibility of reciprocal causality between the carbon productivity in services and some explanatory variables and that it may be affected by some unobservable factors, the existence of endogeneity problems could not be ignored here. Thus, we adopted the dynamic panel GMM model (including the DIF-GMM and SYS-GMM models) to address these problems to the greatest extent. Drawing on the practice of Qamruzzaman (2022) [71] and Zhan (2019) [72], we used the lagged period of *CPS* as an instrumental variable, thereby reducing the endogeneity and ensuring unbiased and consistent estimation results.

Under the DIF-GMM and SYS-GMM models (Table 5), the statistics of AR (1) rejected the null hypothesis, while the statistics of AR (2) were unable to reject the original hypothesis. Both indicated that the error sequence of the model did not have second-order autocorrelation but had first-order autocorrelation. At the same time, the Sargan test statistics also supported the validity of all the instrumental variables used in our research. Thus, the model was reasonably used, and the endogeneity problem was solved. In addition, it can be seen that the explanatory variable, the service industry's carbon productivity, with a lag of one period was beneficial to the current *CPS*, indicating that the carbon productivity in services had the characteristics of significant time continuity and dynamic adjustment. In addition, industrial upgrading still promoted carbon productivity under the condition of endogeneity.

**Table 5.** Endogenous test results.

| Variable | DIF-GMM | SYS-GMM |
|---|---|---|
| L. ln *CPS* | 0.8175 *** | 0.9407 *** |
| | (14.01) | (26.27) |
| ln *IS* | 1.1227 ** | 1.6551 *** |
| | (2.27) | (8.10) |
| Other variables | YES | YES |
| _cons | −1.7731 ** | −2.6927 *** |
| | (−2.21) | (−10.84) |
| AR (1) | −3.1971 | −3.2543 |
| | [0.001] | [0.001] |
| AR (2) | 0.5728 | 0.5532 |
| | [0.567] | [0.580] |
| Sargan | 10.9748 | 21.0413 |
| | [0.531] | [0.690] |

Note: *** and ** denote variables significant at 1% and 5% significance levels, respectively; Values in parentheses are Z-values, and square brackets are *p*-values.

## 6. Further Discussion

### 6.1. Regional Heterogeneity Analysis

The above studies confirmed that industrial upgrading can generally promote China's carbon productivity. However, can this core point of view be applied to local regions of China, given the substantial differences in industrial levels and economic among different regions? In other words, whether the impact is differentiated within different regions of China, this issue needs to be further explored. Taking this into account, our research continued to examine the regional heterogeneity of the influence of industrial upgrading. First, the discussion was divided into eastern and midwest areas. According to the divided two subsamples, we continued to use the SAR model under the fixed effects for regression estimation. At the same time, we also verified the regional heterogeneity analysis by Hu et al. (2019) [73] and introduced the dummy variable, *EAST*, for the eastern region. The core explanatory variable, industrial upgrading, in the eastern region was prominently positive, while the coefficient in the midwest area was not significant. The eastern region had good location conditions and a good economic foundation, with a relatively developed service industry compared with the midwest areas. In addition, various emission-reduction technologies in the industry have become increasingly mature, and the role of promoting low-carbon development of the regional service industry through industrial optimization was even more significant. In the midwest area, the carbon reduction targets could not be supported by strong technology due to the relatively lagging development of the service sector and the low efficiency of energy use. Therefore, it was difficult to achieve effective carbon productivity in the service sector by the transformation of a single regional industry.

In order to continue to investigate the heterogeneity of northern and southern China, we take Qinling-Huaihe as the boundary [74], and then established an SAR model for regression after dividing China into northern and southern regions. The dummy variable, *NORTH*, was also introduced to support the results, and the coefficient of ln *IS* × *NORTH* indicated the difference of influence between the northern and southern regions. In both the northern and southern regions, industrial upgrading could promote carbon productivity in the service industry, while the promotion of upgrading in the northern region was relatively stronger (Columns (4)–(6) in Table 6). This was mainly because the northern region was affected by its unique geographical location and the climatic characteristics of low temperature in winter; thus, the development of services had a greater demand for energy there than in the southern region. Therefore, industrial upgrading will bring a greater increase in carbon productivity in services by way of energy technology optimization. In summary, H2 is confirmed.

**Table 6.** Regional heterogeneity results.

| Variable | (1) | (2) | (3) | (4) | (5) | (6) |
|---|---|---|---|---|---|---|
| ln *IS* | 3.5660 *** | 0.6169 | 0.2041 | 3.2449 ** | 2.0518 *** | 0.6960 |
| | (3.20) | (0.69) | (0.29) | (2.32) | (3.29) | (1.04) |
| ln *IS* × *EAST* | | | 3.4441 *** | | | |
| | | | (4.79) | | | |
| ln *IS* × *NORTH* | | | | | | 4.1691 *** |
| | | | | | | (6.46) |
| Other variables | YES | YES | YES | YES | YES | YES |
| $\rho$ | 0.2860 *** | 0.4258 *** | 0.5247 *** | 0.2898 *** | 0.4354 *** | 0.4458 *** |
| | (3.11) | (3.87) | (6.63) | (2.63) | (4.23) | (5.27) |
| $R^2$ | 0.8196 | 0.4255 | 0.6090 | 0.6438 | 0.6439 | 0.6368 |
| LogL | 122.5091 | 110.1779 | 220.9974 | 63.8805 | 200.8565 | 229.9788 |

Note: *** and ** denote variables significant at 1% and 5% significance levels, respectively; Values in parentheses are Z-values.

### 6.2. Moderating Effect Analysis

The above analysis made it clear that industrial upgrading had a certain role in improving carbon productivity. However, in the current social environment, will the impact be disturbed by other factors, such as the environment and the economy? This issue is worth further discussion and analysis. Based on this, we introduced the interaction term, ln *IS* × ln *ER*, of industrial upgrading and environmental regulation and the interaction term, ln *IS* × ln *ED*, of industrial upgrading and economic development in Equation (3), and we used the SAR model under fixed effects to discuss their interactive effects (Table 7).

**Table 7.** Moderating effect test results.

| Variable | (1) | (2) |
|---|---|---|
| ln *IS* | 1.4387 ** | 1.7665 ** |
| | (2.11) | (2.55) |
| ln *IS* × ln *ER* | 1.2879 *** | |
| | (2.97) | |
| ln *IS* × ln *ED* | | 2.0128 *** |
| | | (4.84) |
| Other variables | YES | YES |
| $\rho$ | 0.5091 *** | 0.3261 *** |
| | (6.23) | (3.17) |
| $R^2$ | 0.6040 | 0.6362 |
| LogL | 214.1701 | 221.6399 |

Note: *** and ** denote variables significant at 1% and 5% significance levels, respectively; Values in parentheses are Z-values.

Table 7 reports the interactive effects of *IS* and *ER*. The regression coefficient is in the same direction as the regression coefficient of industrial upgrading, and the environmental regulation strengthening helps to strengthen the role of industrial upgrading in promoting carbon productivity in services. The "cost effect" exhibited the effect of environmental regulation. However, with the enhancement of environmental regulation, its entry threshold for enterprises in various industries and the resulting "innovation compensation effect" will help expand the effect on the service industry's productivity. Thus, H3 is verified.

The interaction coefficient of 2.0128 between industrial upgrading and economic development was significant. The significance direction was the same as that of industrial upgrading. This showed that industrial upgrading and economic development also had a synergistic effect. It can be roughly understood that with the economy growing, the more active the impact of industrial upgrading will be; thus, H3 is established. The regions with better economic development often have more complete infrastructure constructions; local governments are more capable of subsidizing modern service industries that are dominated by producer services and creating a favorable institutional environment for

them. In addition, Figure 4 shows that the industrial advantages are more pronounced in the higher of the two scenarios compared with the low environmental regulation and economic development levels.

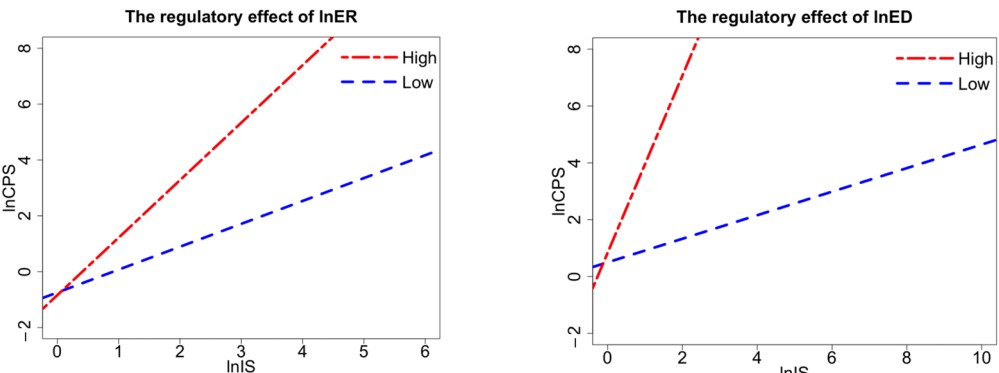

**Figure 4.** Moderating effect diagram of environmental regulation and economic development.

## 7. Conclusions and Limitations

### 7.1. Conclusions

This paper examines the impact characteristics and related mechanisms of industrial upgrading on carbon productivity in the service industry using provincial-level panel data from 2005–2019 in China. The findings are as follows: (1) China's regional service industry's carbon productivity has a strong spatial dependence. (2) Overall, industrial upgrading is beneficial to the carbon productivity in services. (3) From the perspective of the different regions, industrial upgrading has an obvious effect only in the eastern region, while the carbon productivity in the northern area is slightly more affected by industrial upgrading. (4) The results of the moderating effect show that both environmental regulation and economic development enhance the promotion of industrial upgrading on the service industry's carbon productivity. Therefore, we propose the countermeasures of Table 8 based on the above results.

**Table 8.** Directions and tools for reducing carbon productivity in China's service sector.

| Variable | Directions | Tools (Activities) |
|---|---|---|
| Carbon productivity in services | RD/SAE | • Break the constraints of geographical space, promote the integration and reorganization of the production factors in the inter-regional service industry, and require the overall planning to establish a resource allocation system so as to ensure the rationality and fluency of the flow of factor resources.<br>• All regions should learn from each other's important experiences in improving the carbon productivity and actively introduce advanced technology with high efficiency and specialization on the basis of their own advantages of resources, allowing them to form a positive interaction among regions and to reduce the regional differences in carbon productivity. |
| Industrial upgrading | P | • Create a standardized and green industrial environment, gradually eliminate backward enterprises with high emissions, vigorously develop knowledge-intensive and labor-intensive high-tech enterprises, and scientifically and rationally increase the degree of industrial agglomeration so as to drive the realization of a larger-scale economy.<br>• In light of the current service internal structure in various regions, the low-end service industry driven by factors should be guided in an orderly manner to transform it to a high-end modern service industry. The eastern and southern regions can actively introduce emission reduction technologies and improve carbon productivity through the development of renewable energy. The midwestern and northern regions should consider industrial transformation. |

**Table 8.** *Cont.*

| Variable | Directions | Tools (Activities) |
|---|---|---|
| Environmental regulation | N | • All regions should strengthen investment in environmental pollution control, comprehensively improve across-the-board control, form the entry threshold for the service industry market, and generate innovative compensation effects.<br>• The government should improve the environmental regulation strategies. Specifically, enterprises can be controlled by selecting environmental regulation tools based on market incentives, such as the implementation of a carbon emission rights trading system, pollution charging systems, and energy-saving technology subsidy systems.<br>• Relevant departments can strengthen the application for green mark certification and promote enterprises to improve green technology innovation. Meanwhile, consumers are encouraged to buy green services or products, and sustainable consumption is vigorously advocated. |
| Technological progress | P | • The government needs to increase public financial input or policy subsidies to guide enterprises to innovate in traditional technologies and promote new technologies.<br>• By promoting cooperation between enterprises and scientific institutions, it can drive and enhance the low-carbon technological capabilities of the service industry, improve the service sector's carbon productivity, and ultimately effectively stimulate the process of regional green economy. |
| Urbanization | N | • Regions should promote the process of urbanization step by step, emphasis urbanization quality, and improve the technical level of cities and towns so as to avoid the adverse effects of urbanization on the service sector's carbon productivity. |
| Economic development | P | • The country should adhere to the strategy of "development is the first priority" and promote economic development.<br>• Regions with higher economic development levels can focus on reducing carbon dioxide emissions, promote industrial and human capital agglomeration through certain economic incentive policies, and drive low-carbon development of the service industry.<br>• Regions with poor economic development should actively develop their economies, narrow the regional economic gap, and strive to improve production efficiency. |
| Education level | O | / |
| Financial development | O | / |
| Workforce size | P | • Improve the service sector's labor scale, increase the input of high-quality labor factors, and attach importance to the cultivation of high-quality talents. |
| Energy consumption | N | • Optimize the energy consumption structure of the service industry in an orderly manner.<br>• The government can issue relevant policies to increase the utilization of clean energy, reduce the dependence on fossil energy, and realize clean production. |

Note: RD and SAE represent regional differences and spatial agglomeration effects, respectively. P, N, and O denote positive, negative, and no impact, respectively.

*7.2. Limitations*

Admittedly, this study has some limitations that need to be further explored in depth in future research. First, there are many factors affecting the service industry's carbon productivity, and this paper focuses on the factor of industrial upgrading. In the future, we can further explore other key factors of carbon productivity and their influencing mechanisms. Second, although this study selects nine important factors as control variables, it may not be comprehensive enough. Finally, considering the availability of data, this study selects China's inter-provincial data for research. In the future, more powerful evidence can be obtained based on comprehensive prefecture-level city data.

**Author Contributions:** S.W., writing—original draft, conceptualization, methodology, software, formal analysis; J.C., resources, investigation, writing—review and editing, validation. All authors have read and agreed to the published version of the manuscript.

**Funding:** This research was funded by the National Social Science Fund of China, grant number 22BTJ024 and the Natural Science Foundation of Fujian Province, grant number 2020J01170.

**Institutional Review Board Statement:** Not applicable.

**Informed Consent Statement:** Informed consent was obtained from all subjects involved in the study.

**Data Availability Statement:** The data used in the study are available from the corresponding author upon request.

**Conflicts of Interest:** The authors declare that they have no known competing financial interests or personal relationships that could have appeared to influence the work reported in this paper.

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
