# Peer review of "How Does Industrial Upgrading Affect Carbon Productivity in China’s Service Industry?"

_sustainability, doi:10.3390/su151310580_

Round 1
Reviewer 1 Report
1. Clarify the research question: The article addresses the impact of industrial upgrading on carbon productivity in China's service industry. It suggests that this impact can occur through three main mechanisms: adjusting the proportion between industries, promoting technological innovation and energy-saving technologies, and optimizing industrial structure for green development. While the research question is stated clearly, further clarification would be helpful to ensure that readers have a precise understanding of the study's objectives.
2. Provide more context: To enhance the article's comprehensibility, it would be beneficial to provide additional context regarding the broader environmental and economic policy landscape in China. This contextual information would help readers understand how the study contributes to ongoing debates and discussions in the field. Although the article briefly introduces the theoretical background and empirical research on the topic, expanding this section to incorporate more comprehensive references and a discussion of relevant policies would be advantageous. Additionally, providing an overview of the current state of China's service industry, along with its environmental challenges, would improve the article's overall contextualization.
3. Discuss limitations in more detail: The article mentions the use of a spatial autoregressive panel model to investigate carbon productivity in China's service industry. It also outlines the research design and methodology employed to address the research questions and hypotheses. While the article briefly acknowledges limitations, it would be advantageous to discuss these limitations in greater detail and explore their implications for future research. Elaborating on potential methodological constraints, data limitations, or other relevant factors that may have influenced the study's outcomes would enhance the rigor and transparency of the research.
4. Consider alternative explanations: While the article presents empirical evidence supporting the positive effect of industrial upgrading on carbon productivity in China's service industry, it is important to acknowledge the possibility of alternative explanations. Discussing potential confounding factors or alternative causal mechanisms would strengthen the article's argument and demonstrate a thorough consideration of other factors that could contribute to the observed outcomes. By addressing potential counterarguments and offering compelling reasons why industrial upgrading remains the primary driver of carbon productivity improvements, the article can bolster its overall persuasiveness.
5. Provide policy recommendations: The article summarizes the main findings regarding the positive effect of industrial upgrading on carbon productivity in China's service industry, as well as the regional heterogeneity observed in this relationship. It also includes detailed regression results and statistical analysis to substantiate these findings. While the article suggests policy recommendations based on the study's results, it would be valuable to offer more specific and actionable recommendations for policymakers and practitioners. Providing practical guidance, such as specific policy interventions, implementation strategies, or sector-specific measures, would enhance the article's relevance and applicability to real-world decision-making processes.
It appears to be written in a clear and concise manner with appropriate technical terms and statistical analysis. The article is also well-structured with clear headings and subheadings, which makes it easy to follow.
Reviewer 2 Report
I thoroughly reviewed the paper title as: “How Does Industrial Upgrading Affecting Carbon Productivity in China’s Service Industry?” with Manuscript ID: sustainability-2464093. I am pleased to read and review this interesting article. The article is well written and well explained. The authors had presented a detailed literature and theoretical background to support the main topic and to construct the empirical models. Moreover, the empirical part is well established and presented the detailed information and significance of selected models and variables. Overall, the paper is a good contribution to the existing literature. However, I would like to suggest some recommendations to improve the readability of this article for the general readers of this journal. Please explain and make suggested changes as follow:
The introduction part is less established, I would suggest to revise the introduction part and explain the significance main explanatory variable in the introduction part.
Line 49-50# Author believed that energy savings reduction is deepening over the years, however, my concern is that many studies had found that energy savings are improving over the years in China. Moreover, China government is also working closely to improve the energy savings, which are contradictory to the author claimed in the present study, therefore, I would highly recommend authors to please support this claim with some references. Or please explain in detail about this claim refer to line 49-50 that is “Simultaneously, with the deepening of energy-saving reduction, the marginal benefit of reducing emission in industrial sector is decreasing.
Line # 36-38: Author claimed that “The service sector has long been considered a “clean” industry with low energy consumption and emissions, so China’s energy conservation and emission reduction policies mainly focus on traditional energy-intensive industries” However, I would highly recommend to cite the related literature to support this argument. There are plenty of studies which had already been carried on the related topic such as:
1. https://doi.org/10.1016/j.resourpol.2022.102800
2. https://doi.org/10.1016/j.energy.2020.118247
I could not find the clear contributions of this study; Authors need to clearly add the contribution of the present research to the scientific literature in the introduction section.
Reviewer 3 Report
The article is prepared on a current topic, contains econometric calculations and most importantly has all the signs for a recommendation for publication, but there are a number of critical recommendations that must be taken into account for a final positive conclusion.
Thank you!

Round 2
Reviewer 1 Report
Accept in present form
Reviewer 3 Report
Most of the recommendations according to the review have been taken into account, and the choice of variables in the analysis remains debatable. In further studies, consider that the choice of variables should be proven, in particular through correlational methods or others.
The article has been improved and may be recommended for publication.
Thank you!